

# Evidence of rare occurrences of the Phoenix effect in the Hawaiian corals *Porites compressa* and *Montipora capitata* following mortality induced by a marine heatwave

Katelyn G. Jones and Katie L. Barott

Department of Biology, University of Pennsylvania, Philadelphia, PA, United States

Corresponding author
Katie L. Barott,
kbarott@sas.upenn.edu

## ABSTRACT

Coral bleaching is a common stress response to extreme temperatures experienced during marine heatwaves. Bleached corals are left vulnerable without the nutritional support of their algal symbionts, and can often suffer partial or complete mortality. Bleaching-induced mortality is often accompanied by colonization of turf algae over the dead coral skeleton, which can be difficult for corals to regrow over. The Phoenix effect is a phenomenon of rapid recovery of live coral tissue following mortality, which is hypothesized to occur *via* the regrowth of tissue from deep within the coral skeleton that expands over the top of dead portions. Here, we found that the Hawaiian corals *Porites compressa* and *Montipora capitata* can display rapid tissue recovery suggestive of the Phoenix effect. During a marine heatwave that occurred in 2015 in Kāne'ohe Bay, Hawai'i, USA, 237 individuals (including bleached and non-bleached phenotypes) were identified and monitored for mortality and recovery over the next 2–7 years. Nearly 16% of *P. compressa* individuals and 34% of *M. capitata* exhibited substantial partial mortality, and approximately half of these affected individuals of each species had bleached during the heatwave. Partial mortality following the 2015 heatwave was followed by turf algae colonization over the exposed skeleton. Of the colonies with substantial mortality, six colonies (10% of affected individuals; five *P. compressa* and one *M. capitata*) subsequently recovered to over 90% live coral tissue within 2 years (2017), with an additional three colonies (two *P. compressa* and one *M. capitata*) recovering within 4 years of the 2015 marine heatwave (2019). We qualify colonies with rapid tissue recovery as those that meet two criteria: (1) substantial partial mortality (≥40%) in the first 12 months following the initial 2015 marine heatwave, and (2) recovery of any amount of live tissue at anytime before 2022. Interestingly, only colonies that had bleached in 2015 exhibited rapid tissue recovery. A consecutive, yet less severe marine heatwave occurred in 2019, and none of the previously recovered colonies observed experienced significant tissue loss, suggesting these individuals remained resilient amidst a secondary heat stress exposure. This phenomenon is an example of remarkable recovery and resilience that may be informative for further study of mechanisms of coral tissue regeneration in two important reef-building coral species.

## INTRODUCTION

Anthropogenic greenhouse gas emissions have caused an increase in the frequency and severity of marine heatwaves since the early 19th century (*Oliver et al., 2019*, *2021*). Coral reef ecosystems are particularly threatened by recurring marine heatwaves, as heat stress is a primary driver of mass coral mortality events (*Brown, 1997*; *Hughes et al., 2017*; *Van Woesik et al., 2022*). Even under moderate global emissions scenarios that limit future warming to 1.5 °C, ≥70% of global coral reef populations are predicted to decrease as marine heatwaves are projected to correspondingly increase in the coming decades (*Oliver et al., 2019*; Fig. SPM.4c, *IPCC, 2023*). Heat stress induces coral bleaching, in which colonies transition from fully-pigmented tissues containing high densities of dinoflagellates (family Symbiodinaecae) to "bleached," transparent tissues resulting from the loss of their symbionts (*Putnam et al., 2017*; *Van Woesik et al., 2022*). Symbiotic dinoflagellates are a fundamental component of the holobiont, providing metabolites to the coral host (*Davy, Allemand & Weis, 2012*). Bleached corals are deprived of crucial nutrition due to the disruption to their obligate symbiosis, which can result in death if algal populations are unable to recolonize host tissues (*Putnam et al., 2017*). Partial mortality of the colony can be followed by rapid colonization (within 3 months) of turf algae over the newly exposed skeleton (*Diaz-Pulido & McCook, 2002*; *Diaz-Pulido et al., 2009*). Not only does turf algal colonization reduce the integrity of individual colonies, but it also may lead to a moderate or severe phase-shift on the reef through dominance of macroalgae (*Bruno et al., 2009*; *Bahr, Jokiel & Toonen, 2015*; *Putnam et al., 2017*).

Importantly, some corals have shown rapid rates of tissue regrowth following bleaching-induced mortality that exceed predicted growth rates of colonization of new substrates (*Diaz-Pulido et al., 2009*; *Roff et al., 2014*; *Sheppard & Sheppard, 2020*). For example, recovery in massive *Porites* spp. is expected to take as much as 100 years due to their slow growth rates (*Mumby et al., 2001*), but *Roff et al. (2014)* found large massive *Porites* colonies that suffered extensive mortality following a bleaching event in 1998 (Rangiroa Atoll, French Polynesia) had recovered in just 15 years. This accelerated recovery, specifically describing the regrowth of live tissue over previously dead skeleton, has been termed the "Phoenix effect" (PE; *Krupp, Jokiel & Chartrand, 1993*). The mechanism of this rapid recovery that results in the PE is currently unknown, but the regrowth of cryptic tissue remaining deep within the skeleton of the colony and/or from adjacent areas live tissue (*i.e.*, resheeting) is a leading hypothesis (*Franzisket & für Naturkunde, 1970*; *Krupp, Jokiel & Chartrand, 1993*; *Riegl & Piller, 2001*; *Roff et al., 2014*; *DeFilippo et al., 2016*; *Furby, 2017*). In addition to massive *Porites* spp. (*Roff et al., 2014*), a diversity of coral species have been observed to recover *via* the PE, including the solitary coral *Lobactis* (formerly *Fungia*) *scutaria* (*Krupp, Jokiel & Chartrand, 1993*), the corals *Orbicella* (formerly *Montastrea*) *annularis* (*Meesters, Noordeloos & Bak, 1994*), Faviids and *Ctenella* spp. (*Sheppard & Sheppard, 2020*), *Acropora* spp. (*Diaz-Pulido et al., 2009*),

*P. astreoides* (Meesters et al., 1992) and *P. compressa* (*Jokiel et al., 1993*), and the temperate coral *Astrangia poculata* (*DeFilippo et al., 2016*). The PE may thus represent an important mechanism of resilience for colonies that have experienced mortality and may become critical for reef recovery as marine heatwaves and bleaching events increase in frequency and severity on coral reef ecosystems around the globe (*Van Woesik et al., 2022*). On the individual level, rapid recovery *via* the PE could prevent partially dead colonies from being outcompeted for space on the benthos. Additionally, the existence of resilient individuals is crucial amidst continual climate warming and environmental disturbances, and species that display the PE might represent resilient populations that can preserve reef structure at large (*Hughes et al., 2017*). For reef ecosystems that have been under prolonged periods of stress from anthropogenic disturbances, species that show rapid recovery in local populations might facilitate the persistence of the overall reef structure.

One such crucial, yet vulnerable, collection of reefs is located in Kāne'ohe Bay, Hawai'i, USA. Historically, the coral reefs in Kāne'ohe Bay have experienced both direct and indirect consequences of human action through major disturbances including dredging, sewage input, freshwater reef kills, and marine heatwaves (*Jokiel et al., 1993*; *Bahr, Jokiel & Toonen, 2015*). This well-described system is still under environmental pressure despite actions to decrease direct harm to the bay (*Bahr, Jokiel & Toonen, 2015*). For example, coral populations in Kāne'ohe Bay have already survived four mass bleaching events, which occurred in 1996, 2014, 2015, and 2019 (*Jokiel & Brown, 2004*; *Bahr, Rodgers & Jokiel, 2017*; *Innis et al., 2021*; *Yadav et al., 2023*). Individuals of the two dominant stony coral species in Kāne'ohe Bay, *Porites compressa* and *Monitpora capitata*, have shown divergent phenotypic responses to these marine heatwaves, where some colonies bleach while other colonies of the same species remain fully pigmented (*Matsuda et al., 2020*; *Ritson-Williams & Gates, 2020*; *Brown et al., 2023*; *Yadav et al., 2023*). After the repeat bleaching events in 2014 and 2015, bleaching-susceptible and bleaching-resistant colonies of each species were tagged for ongoing investigation (*Matsuda et al., 2020*; *Brown et al., 2023*). *Matsuda et al. (2020)* found that bleaching-susceptible colonies of *P. compressa* showed greater partial mortality than bleaching-resistant colonies following the 2015 heatwave, but recovered rapidly from bleaching and on average sustained low rates of partial mortality (<20%; *Matsuda et al., 2020*). In contrast, both phenotypes of *M. capitata* suffered substantial partial mortality, with bleaching-susceptible corals experiencing greater losses of live tissue than bleaching-resistant individuals (*Matsuda et al., 2020*). Here, we sought to determine whether any *P. compressa* or *M. capitata* colonies that suffered significant partial mortality following the 2015 marine heatwave displayed rapid recovery of live tissue *via* the PE. In addition, we sought to determine how common the PE is in these two species and the rate at which this recovery occurs. To do so, we queried the mortality and recovery dynamics of 237 colonies (176 previously described by *Matsuda et al. (2020)* and 61 new colonies) of *P. compressa* and *M. capitata* in Kāne'ohe Bay, Hawai'i from 2015–2022, which encompassed two marine heatwaves and associated coral bleaching events (2015 and 2019), for examples of substantial partial mortality followed by tissue regrowth. This study will potentially identify resilient individuals within coral reef populations that demonstrate the ability to rapidly recover from bleaching-induced

mortality, which would provide a mechanism to preserve and recover reef structure in ecosystems such as Kāne'ohe Bay.

## MATERIALS AND METHODS

### Identification of corals potentially exhibiting the Phoenix effect

Bleached and non-bleached colonies of *Porites compressa* (*n* = 119) and *Montipora capitata* (*n* = 118) were tagged *in situ* during a marine heatwave that occurred in Kāne'ohe Bay, Hawai'i in late 2015. A total of 22 bleached and 42 non-bleached colonies of each species were tagged at patch reef 4 (21.4339, −157.7984). In addition, a total of 22 bleached and 54 non-bleached corals were tagged at patch reef 13 (21.4509, −157.7954) (Table 1; *Matsuda et al., 2020*). Both patch reefs are located within the lagoon of Kāne'ohe Bay, and patch reef 4 is 0.75 km from shore with a ~30 day seawater residence time while patch reef 13 is 1.6 km from shore with a ~1 day seawater residence time (*Lowe et al., 2009*). Each bleached colony was located adjacent to a conspecific non-bleached colony, forming 22 phenotype pairs of each species at each site. All colonies were between 1–2 m deep. The bleaching and recovery responses of these pairs have been described in *Matsuda et al. (2020)*, *Innis et al. (2021)*, and *Brown et al. (2023)*. Photographs of each colony were taken from 2015–2017 (patch reefs 4 and 13; 2–3 month intervals) and 2019–2022 (patch reef 13 only; ~6 month intervals). Partial mortality of each colony was visually assessed in 20% intervals as described in *Matsuda et al. (2020)*. Partial mortality of the pairs of bleached and non-bleached colonies is published in *Matsuda et al., 2020* (data from 2015–2017) and *Brown et al., 2023* (data from 2019–2022). Partial mortality of the unpaired non-bleached colonies is reported here for the first time (*n* = 61). Candidate colonies for the Phoenix effect (PE) were identified based on these partial mortality assessments if they met the following two criteria: 1) exhibited substantial partial mortality (≥40%) in the first 12 months following the 2015 marine heatwave, and 2) recovered any amount of live tissue at anytime before 2022. Images of these colonies were then analyzed for a more precise quantification of partial mortality as described below. Tagging and monitoring of corals was conducted with the approval of the State of Hawaii Department of Land and Natural Resources, Office of Conservation and Coastal Lands, under Site Plan Approval permit No. OA-16-25.

### Precise quantification of colony partial mortality

For each candidate colony meeting the two criteria described above (*n* = 9), the area of live tissue covering the surface of each colony at each available time point was quantified in ImageJ, version 2.14.0/1.54f (*Schindelin et al., 2012*). Specifically, for each image the Freehand Selection tool was used to outline the entire colony, and this area was saved as a region of interest (ROI). If the colony was obstructed by field tools (color cards, rulers, and tags) or fish, these objects were also saved as an ROI. The area of the colony was then measured (Total Area) by subtracting the area of the obstructing item ROIs from the colony ROI. Any subsections of the colony that appeared dead were then each outlined and saved as individual ROIs. Dead skeleton was indicated by a textured, non-uniform green surface (usually turf algae) that contrasted with the smooth texture of both healthy and

**Table 1 Total number of colonies of each bleaching phenotype by species and location at each step of the analysis (rows).** Colonies exhibiting substantial partial mortality (≥40%) as quantified *via* visual estimates within 1 year of the 2015 marine heatwave (second row) that also showed recovery of live tissue at any time up through 2022 (last row) were selected for precise quantification of partial mortality in ImageJ. ND, no data.

| | Porites compressa | | | | Montipora capitata | | | |
| --- | --- | --- | --- | --- | --- | --- | --- | --- |
| | Patch Reef 4 | | Patch Reef 13 | | Patch Reef 4 | | Patch Reef 13 | |
| | Bleached | Pigmented | Bleached | Pigmented | Bleached | Pigmented | Bleached | Pigmented |
| Colonies tagged during the 2015 MHW | 22 | 42 | 22 | 33 | 22 | 42 | 22 | 32 |
| Colonies with substantial partial mortality (≥40%) within 1 year of the 2015 MHW | 6 | 5 | 7 | 1 | 18 | 17 | 3 | 2 |
| Colonies showing recovery of live tissue by 2017 | 0 | 0 | 5 | 0 | 0 | 0 | 1 | 0 |
| Colonies showing recovery of live tissue by 2022 | ND | ND | 7 | 0 | ND | ND | 2 | 0 |

bleached live coral tissue with visible polyps (Fig. 1, inset). Sponges visible on the surface of the colony were also categorized as dead colony portions. The area of all dead skeleton ROIs were summed to get a measure of total partial mortality across the colony (Dead Area). Percent partial mortality of each colony was calculated by the ratio of Dead Area: Total Area and multiplying by 100. Percent live tissue of each colony was calculated by subtracting the total percent partial mortality from 100%. ROIs of live tissue, dead skeleton, and obstructing objects were shaded using the Overlay feature in ImageJ for a composite visualization of colony health (Fig. 1). A minimum of 7 and a maximum of 16 time points were scored for each individual. Some colonies were missing time points as a result of either difficulties locating individuals in the field or exclusion of images with incomplete coverage of the colony from the analysis (ND-no data, Table 1). Colony diameter was measured in the field using a transect tape.

### Examination of live tissue recovery dynamics

Recovery time for each colony was determined by the number of months between the time point with the greatest extent of partial mortality to the first time point where significant recovery to ≥90% live tissue was achieved. The influence of initial colony size and the maximum extent of mortality on recovery time were explored using a linear model. All statistical analyses and graphs were created using R version 4.2.2 (2022-10-31) (*R Core Team, 2022*).

## RESULTS

### Patterns of coral bleaching, tissue loss and recovery following the 2015 marine heatwave

A total of 59 out of 237 total colonies observed experienced substantial mortality (>20% tissue loss) within 1 year of the 2015 marine heat wave (Table 1). Of these, 53 (89.8%) showed no signs of live tissue recovery within 24 months. For *P. compressa*, 19 of the 119 total colonies that were followed (16%) exhibited substantial partial mortality following the heatwave (Table 1). Of these, 13 had bleached in 2015 while the other six had not, and they were split about evenly between the two reef locations (Table 1). Only five of the 19

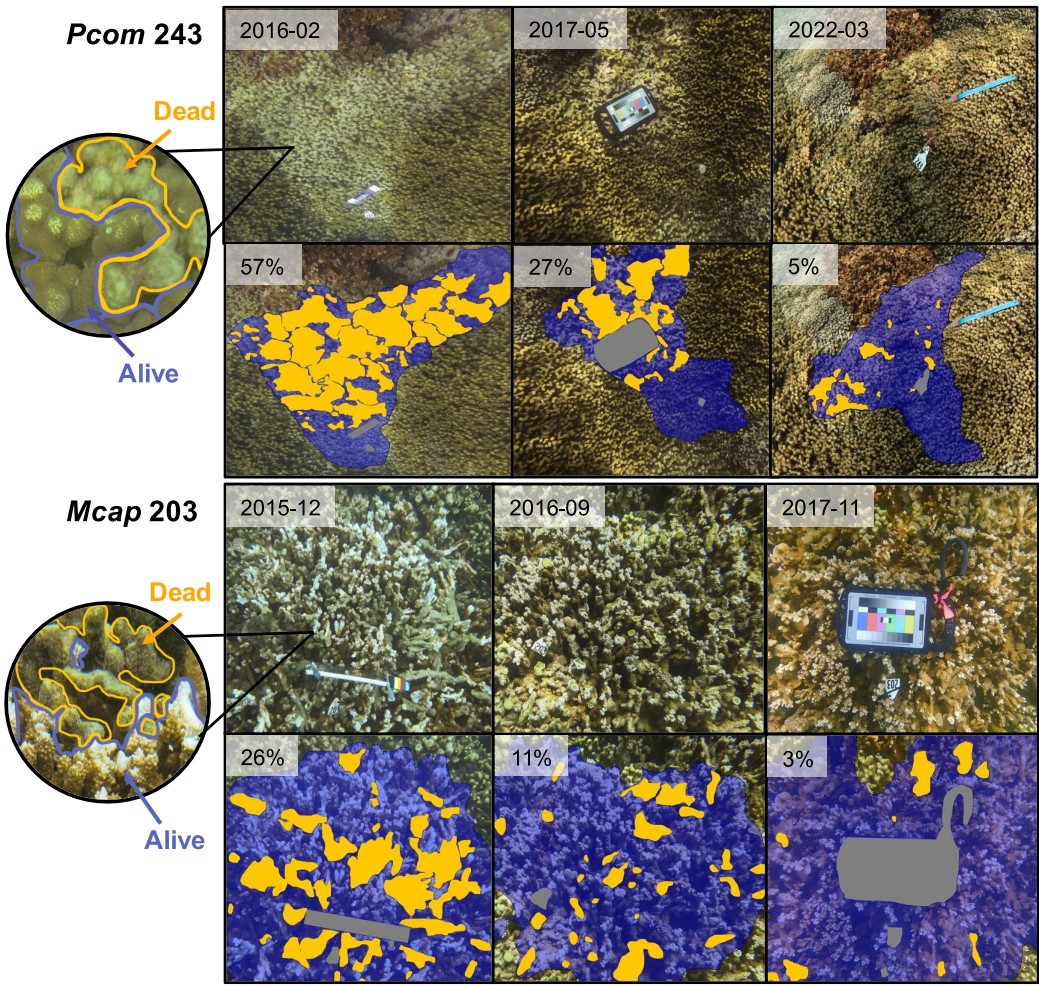

**Figure 1 Images of individual coral colonies displaying significant recovery of live tissue following bleaching-induced mortality.** Each image set contains original photographs of a single colony over time on the top row, followed by the same image on the bottom row with an overlay indicating portions of live tissue (blue), dead tissue (orange), and obstructing objects (gray). Circle image shows a close-up image of each colony with live and dead portions outlined in the corresponding colors and indicated by arrows. Dead portions of the colony were identified by colonization of turf algae, as indicated by increased texture and discoloration. Inset within each image indicates date of image (top row) or total partial mortality (bottom row). Row labels indicate colony ID and species (Pcom, *Porites compressa*; Mcap, *Montipora capitata*).

*P. compressa* colonies that suffered substantial mortality showed evidence of live tissue recovery within 2 years of the 2015 marine heatwave and another two colonies exhibited recovery within 4 years, for a total of seven colonies out of the 19 (37%) exhibiting recovery at sometime before 2022 (Table 1). In addition, all of the *P. compressa* colonies exhibiting recovery had bleached during the 2015 heatwave, and all were located at patch reef 13 (Table 1). The mean partial mortality of the other 14 *P. compressa* colonies that did not recover live tissue within 2 years after the 2015 marine heatwave was 57%. For *M. capitata*, 40 of the 118 colonies that were followed (34%) exhibited substantial mortality within 1 year of the 2015 marine heatwave, and just over half of these colonies (21/40) had bleached
**Table 2 Individual colony metadata of corals exhibiting substantial partial mortality and subsequent recovery of live tissue following the 2015 marine heatwave.** Bleaching severity was defined as none (0%), mild (<20% colony), moderate (20–50% colony), moderately severe (50–80% colony), or severe (>80% of colony white). Time to recovery indicates the time elapsed between the maximum extent of mortality to the first instance of ≤10% partial mortality. ND, no data. Recovery rate is a rough estimate of the amount of surface area recovered over time (months), calculated by dividing the maximum surface mortality (%) by the total recovery time (months).

| | *Porites compressa* | | | | | | | *Montipora capitata* | |
| --- | --- | --- | --- | --- | --- | --- | --- | --- | --- |
| | Colony 43 | Colony 45 | Colony 225 | Colony 241 | Colony 243 | Colony 245 | Colony 247 | Colony 1 | Colony 203 |
| Colony diameter (cm) | 70 | 150 | 160 | 210 | 170 | 100 | 180 | 110 | 75 |
| Bleaching severity 2015[1] | Severe | Severe | Severe | Severe | Severe | Severe | Severe | Severe | Severe |
| Bleaching severity 2019[2] | None | Mild | ND | ND | Mild | ND | Mild | ND | Moderate |
| Maximum mortality (% surface area) | 11 | 55 | 34 | 48 | 57 | 44 | 22 | 15 | 36 |
| Recovery time (months) | 3 | 37 | 9 | 17 | 40 | 17 | 15 | 2 | 15 |
| Recovery rate (% month$^{-1}$) | 3.67 | 1.49 | 3.78 | 2.82 | 1.36 | 2.59 | 1.47 | 7.50 | 2.40 |

**Notes:**
[1] Data from *Matsuda et al. (2020)*.
[2] Data from *Brown et al. (2023)*.

(Table 1). In addition, most of these colonies with substantial partial mortality (35/40; 88%) were located at patch reef 4 (Table 1). However, in contrast to *P. compressa*, only one colony of *M. capitata* that suffered substantial partial mortality showed signs of live tissue recovery by 2017, with one additional colony showing recovery by 2022 (2/40 (5%) of all colonies showing substantial partial mortality) (Table 1). The mean partial mortality of the 38 *M. capitata* colonies that did not recover live tissue within 2 years after the 2015 marine heatwave was 67.2%, and seven of these colonies had died back completely (100% mortality) by 2017. Similarly to *P. compressa*, both colonies of *M. capitata* that showed recovery of live tissue had previously bleached and were located at patch reef 13 (Table 1).

**Tissue loss and recovery dynamics of candidate Phoenix effect colonies**

The detailed quantification of partial mortality correlated positively with the initial less precise visual mortality assessments from *Matsuda et al. (2020)* and *Brown et al. (2023)* ($R^2 = 0.674$), the latter of which tended to overestimate the extent of partial mortality. The detailed mortality quantification carried out in this study revealed the extent and timing of live tissue recovery following substantial tissue loss among individual colonies. The maximum extent of partial mortality for the seven colonies of *P. compressa* ranged from 11–57%, with an average of 39% (±6.3%) (Table 2). The dead regions of each colony were colonized primarily by a mixed community of turf algae (Figs. 1, inset). The time elapsed between the greatest extent of partial mortality to recovery above 90% live tissue took between 9–40 months, with an average recovery time of 22.5 months (±5.2 months; Table 2), and all seven of the *P. compressa* colonies had recovered to over 93% live tissue by 2019 (Fig. 2). The estimated recovery rate ranged from 1.36 to 3.78 percent per month for *P. compressa* colonies (Table 2), however this metric is purely an estimation as recovery

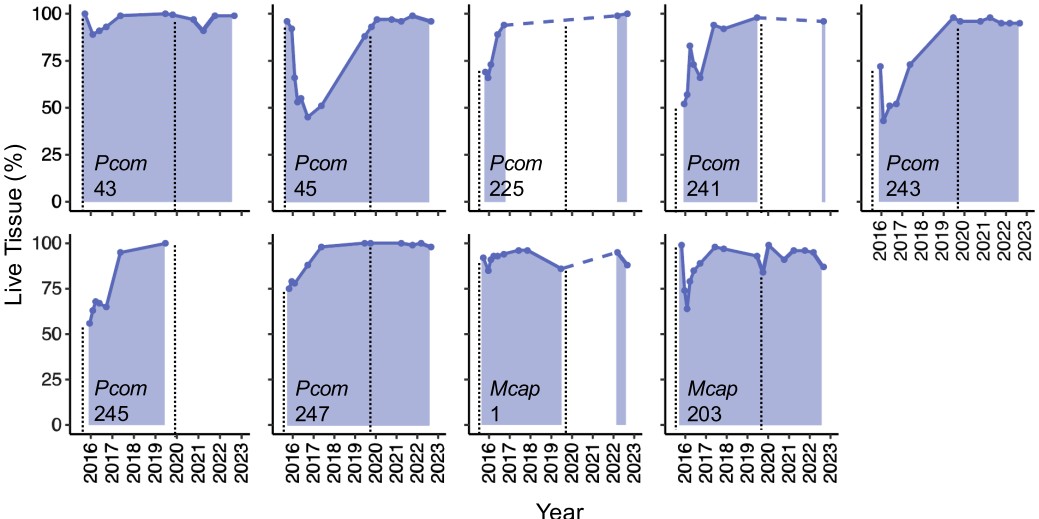

**Figure 2 Patterns of live tissue change for individual *Porites compressa* and *Montipora capitata* colonies exhibiting the Phoenix effect following a severe marine heatwave in 2015.** Live tissue (%) for each colony over time from 2015–2022 (shaded). Inset indicates species (Pcom, *Porites compressa*; Mcap, *Montipora capitata*) and colony ID. Black vertical dashed lines indicate the occurrence of a marine heatwave.

may be more variable over time than these values represent. While all of these colonies had bleached severely in 2015, none of those observed during the 2019 bleaching event (four of seven) showed severe bleaching, with colony 43 exhibiting no signs of bleaching and the other three colonies exhibiting mild paling (Table 2). In the year following the 2019 marine heatwave, these colonies suffered little to no partial mortality (Fig. 2). Unfortunately, there were no observations of four of the seven *P. compressa* colonies within the first year of the 2019 marine heatwave, and so the mortality and recovery patterns following this second heatwave are unknown for these individuals. However, all four colonies had nearly 100% live tissue when they were next observed in 2020 (colony 243), 2021 (colony 247) and 2022 (colonies 225 and 241) (Fig. 2).

The maximum extent of partial mortality for the two individuals of *M. capitata* exhibiting recovery was 15% for colony 1 and 36% for colony 203 (Table 2, Fig. 2). The dead regions of each colony were colonized primarily by a mixed community of turf algae and cyanobacteria (Fig. 1). The time for each colony to recover to ≤10% mortality differed, ranging from 2 months for Colony 1 to 15 months for Colony 203 (Table 2). Following the second heatwave in 2019, Colony 203 underwent moderate bleaching but did not suffer any additional mortality (Fig. 2). There were no observations of Colony 1 during the 2019 heatwave or the year after, so it is not known how this colony responded to that heatwave. However, this colony was located again in 2022, at which point the colony had nearly 100% live tissue coverage (Fig. 2). Overall, there was a significant effect of the maximum extent of mortality on the time it took to recover to ≤10% mortality for both species ($R^2 = 0.764$, $p = 0.008$), such that corals with greater mortality took longer to recover. However, the size of the colonies, which ranged between 70–210 cm maximum

diameter (Table 2), did not have a significant effect on recovery time ($p = 0.944$). There was no significant interaction between the maximum extent of mortality and colony size on recovery time ($p = 0.913$).

## DISCUSSION

Nine individuals of the Hawaiian corals *Porites compressa* and *Montipora capitata* were observed to rapidly recover from substantial partial mortality following a severe marine heatwave that occurred in 2015. All of these individuals recovered to nearly 100% live tissue within a few months to just under 4 years following the heatwave. As these corals were large, ranging in diameter from nearly 1 m to over 2 m, recovery of such a large surface area would likely take decades to achieve through other methods such as either lateral incursion from the adjacent living tissue of the colony or *via* proliferation of new recruits. These results therefore indicate that these corals underwent recovery *via* the Phoenix effect (PE), a recovery process hypothesized to be driven by tissue remaining deep within the skeleton and/or adjacent live tissue remnants along the surface of the colony (*Krupp, Jokiel & Chartrand, 1993*; *Diaz-Pulido et al., 2009*; *Roff et al., 2014*). Both *P. compressa* and *M. capitata* have relatively fast calcification rates (*Rodrigues & Grottoli, 2006*; *Bahr, Jokiel & Rodgers, 2016*), however, other fast-growing species (*e.g.*, *Acropora*) have previously been documented to display this type of rapid tissue regrowth over algal-colonized dead coral skeleton (*Diaz-Pulido et al., 2009*). Interestingly, all colonies that exhibited the PE had bleached during the 2015 marine heatwave, even though a similar number of non-bleached individuals underwent substantial partial mortality. All colonies that recovered *via* the PE avoided further tissue loss following another marine heatwave in 2019. This is likely because the bleaching severity for all of these individuals was much less (mild to moderate) than in 2015, when all colonies had bleached severely. This finding indicates that the rapid gains of live tissue following the first heatwave had meaningful implications for the overall health and survival of these colonies in the long-term, but could also be due in part to the lower severity of the 2019 marine heatwave (5.1 *vs.* 8.8 degree heating weeks, respectively; *Brown et al., 2023*). Interestingly, far fewer *M. capitata* were observed to exhibit the PE than *P. compressa*, despite *M. capitata* having a higher frequency of individuals with substantial partial mortality and thus opportunities to undergo rapid recovery. *M. capitata* appears to be more sensitive to heat stress in this system, as individuals that have been followed since the 2015 marine heatwave in Kāneʻohe Bay have been undergoing a continued increase in partial mortality, including both previously bleached and non-bleached individuals (*Matsuda et al., 2020*; *Brown et al., 2023*). The two *M. capitata* individuals that did recover *via* the PE were thus the exception rather than the norm. Surprisingly, none of the bleaching-resistant individuals (*i.e.*, those that remained pigmented during the 2015 marine heatwave) that suffered substantial mortality recovered live tissue within 2 years of the heatwave. Corals that resist bleaching during a marine heatwave are typically considered more stress resilient than their bleached counterparts, yet we found that these corals still underwent substantial partial mortality following the marine heatwave and none of the non-bleached individuals that suffered substantial partial mortality recovered within 7 years of the 2015 marine heatwave despite

being over-represented relative to bleached individuals in our dataset. These results underscore the severe physiological stress incurred by corals following exposure to anomalously high temperatures, even for corals that do not undergo bleaching (*i.e.*, symbiont and/or pigment loss).

All of the individuals that showed the PE were located on patch reef 13, even though a majority of corals experiencing substantial mortality were located at patch reef 4. These results indicate that there may have been a strong environmental influence on coral recovery following mortality. There are multiple factors that could have led to these different mortality and recovery patterns at these two reef locations. First, while these two reefs are less than 1 km apart, patch reef 4 experienced greater heat stress than patch reef 13 during the 2015 marine heatwave (*Matsuda et al., 2020*). In addition, a variety of environmental factors differ between the two reefs, driven by patch reef 4's location near shore *versus* patch reef 13's location farther from shore within the lagoon near the backreef. As such, patch reef 4 experiences greater terrestrial input, including sewage and freshwater runoff (*Bahr, Jokiel & Toonen, 2015*; *Barott et al., 2021*), as well as longer seawater residence times (*Lowe et al., 2009*) and slower flow rates than patch reef 13 (*Barott et al., 2021*). This difference in the frequency of the Phoenix effect between these two different patch reefs within the lagoon of Kāneʻohe Bay suggests that environmental conditions at each location likely had a significant impact on whether and how quickly corals were able to undergo recovery following mortality from a marine heatwave. Indeed, these patterns were mirrored across the entire population of corals followed, as both bleached and non-bleached individuals of both species showed gradually increasing accumulation of partial mortality at patch reef 4 and patch reef 13 within the first 2 years following the 2015 marine heatwave, with the exception of non-bleached *P. compressa* at patch reef 13 that suffered little to no mortality (*Matsuda et al., 2020*). Interestingly, previously bleached individuals accumulated more mortality on average than corals that had not bleached for both species (*Matsuda et al., 2020*), as might be expected for corals that underwent the stress of bleaching, and yet bleached corals were the only ones observed here to recover live tissue *via* the PE, despite many non-bleached individuals of each species at each location exhibiting substantial partial mortality. This could be an artifact of the small sample size of corals exhibiting the PE, but a better understanding of the mechanism of tissue regrowth would help to determine if bleaching plays a role in this resilience mechanism.

The mechanisms underlying the PE recovery process remain poorly understood but have been hypothesized to occur *via* the growth of tissue that remains alive deep within the coral skeleton that is able to proliferate and "resheet" over the dead skeleton (*Krupp, Jokiel & Chartrand, 1993*; *Roff et al., 2014*; *Furby, 2017*). Perforate corals, which have tissue that penetrates deep crevices and maintains connections between polyps within the skeleton, are expected to be more likely to recover in this manner. Both *P. compressa* and *M. capitata* have perforate skeletons with intercalated tissue that penetrates the skeletons. However, the lower frequency of live tissue recovery *via* the PE in *M. capitata* suggests other factors may impact an individual's ability to undergo this type of recovery, and further study is

necessary to determine which biological factors promote high rates of tissue recovery. For example, if skeletal porosity is indeed necessary to the ability for colonies to show this type of recovery, individuals of species with low-porosity (imperforate) skeletons would not be expected to display the PE. It is clear, however, that this skeletal structure is not sufficient to cause the PE following mortality.

Resheeting over existing skeleton has been hypothesized to be less energetically costly, as less new skeleton is required to be laid down by recolonizing tissues (*Sheppard & Sheppard, 2020*). This might explain why recovery *via* the PE can occur so quickly. However, as was observed here, resheeting often occurs after the skeleton has been colonized by turf algae (*Diaz-Pulido et al., 2009*; *Roff et al., 2014*). The impact of turf algae colonization of dead coral skeletons on the ability of adjacent or residual coral tissues deeper within the skeleton to recolonize the skeletal surface are not well understood, but competition with turf algae likely hinders coral tissue regrowth. For example, turf algae competition with coral often incurs negative consequences for the coral, although the mechanisms vary across specific coral-algal species interactions and environmental conditions (*Jompa & McCook, 2003*; *Bender, Diaz-Pulido & Dove, 2012*; *Barott & Rohwer, 2012*; *O'Brien & Scheibling, 2018*). Filamentous turf algae can reduce tissue recovery and regrowth following the creation of lesions (*Bender, Diaz-Pulido & Dove, 2012*), potentially through competition for resources, overgrowth of microbial organisms, or greater recruitment of pathogens (*Barott et al., 2012*; *O'Brien & Scheibling, 2018*; *Roth et al., 2018*). In addition, turf algae coverage on the reef benthos is often associated with a degraded reef structure (*Barott & Rohwer, 2012*; *O'Brien & Scheibling, 2018*), and once a reef becomes dominated by benthic algae, recovery of reef-building corals often requires a timescale of decades (*McManus & Polsenberg, 2004*). An increasing number of other environmental disturbances, including eutrophication, overfishing and climate change are predicted to prolong this recovery time (*Hughes et al., 2017*). The corals in Kāne'ohe Bay have now experienced three successive marine heatwaves in the last decade (2014, 2015, and 2019) (*Bahr, Jokiel & Toonen, 2015*; *Brown et al., 2023*), yet the ability observed here for rapid recovery from mortality and turf algal colonization in nine individuals of two dominant coral species provides a ray of hope for reef recovery in a changing climate.

## CONCLUSIONS

Here we found that a small number of individuals of the coral species *Porites compressa* and *Montipora capitata* were able to rapidly regrow live tissue over dead skeleton following tissue loss in the aftermath of a severe marine heatwave. The biological mechanisms that allow for this rapid recovery, also known as the Phoenix effect, require further study and would provide valuable information about how coral recovery occurs on a micro-scale. Corals that show rapid tissue regeneration following partial colony mortality can be used to understand the divergence between species and/or individuals within a species with distinct recovery responses. For example, it is unknown whether the capacity for this recovery mechanism is primarily genetically determined, made possibly due to the underlying skeletal and surviving tissue architecture, or a combination of these and the

surrounding environmental conditions on the reef. Understanding natural mechanisms of coral tissue recovery may be useful for informing policies to protect vulnerable reef systems with species that are likely to recover rapidly, as species that are able to recover quickly from mortality may help promote coral reef stability and ecosystem function. Bleaching-susceptible coral colonies that are able to recover rapidly, rather than becoming further sensitized to heat stress, may be the most likely to dominate reefs in the coming decades (*Brown & Barott, 2022*). It is important to note that only a small proportion of individuals that suffered substantial mortality exhibited rapid recovery of live tissue, and the loss of live tissue from coral colonies that did not recover is likely to lead to declining structural complexity and depressed growth of the reef substratum as marine heatwaves repeatedly impact these ecosystems. The PE may be one mechanism of resilience that allows for some individuals to survive through multiple marine heatwaves. Overall, better understanding the prevalence and mechanisms of this phenomena could lead to gains in knowledge about the capacity for reef recovery as more marine heatwaves and accompanying losses in live coral cover are anticipated in the coming decades.

## ACKNOWLEDGEMENTS

The authors would like to thank Linda Wu for image organization and Dr. Kristen T. Brown and Benjamin Glass for helpful insights and guidance on data analyses and visualization. We also thank Crawford Drury and the Hawaii Institute of Marine Biology for logistical support.

### Funding

This work was supported by the National Science Foundation (No. 2237658 and No. 1923743) to Katie L. Barott. The funders had no role in study design, data collection and analysis, decision to publish, or preparation of the manuscript.

### Grant Disclosures

The following grant information was disclosed by the authors:
National Science Foundation: 2237658 and 1923743.

### Competing Interests

The authors declare that they have no competing interests.

### Author Contributions

- Katelyn G. Jones conceived and designed the experiments, performed the experiments, analyzed the data, prepared figures and/or tables, authored or reviewed drafts of the article, and approved the final draft.
- Katie L. Barott conceived and designed the experiments, performed the experiments, analyzed the data, prepared figures and/or tables, authored or reviewed drafts of the article, and approved the final draft.

### Field Study Permissions

The following information was supplied relating to field study approvals (*i.e.*, approving body and any reference numbers):

Tagging and monitoring of corals was conducted with the approval of the State of Hawaii, Department of Land and Natural Resources, Office of Conservation and Coastal Lands, under Site Plan Approval permit No. OA-16-25.

### Data Availability

The data and code are available on Dryad at: Jones, Katelyn; Barott, Katie (2025). Data and code from: Evidence of rare occurrences of the Phoenix effect in the Hawaiian corals *Porites compressa* and *Montipora capitata* following mortality induced by a marine heatwave [Dataset]. Dryad. https://doi.org/10.5061/dryad.p2ngf1w2j.

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
