# Peer review of "Evidence of rare occurrences of the Phoenix effect in the Hawaiian corals Porites compressa and Montipora capitata following mortality induced by a marine heatwave"

_PeerJ, doi:10.7717/peerj.19225_

## Round 0.1 · original submission · Major Revisions

Two expert reviewers have evaluated your manuscript and their detailed comments and insightful suggestions tocan be seen below. Please ensure that you address all of these comments and suggestions in a point by point response that clearly indicates where modifications have been made in the revised manuscript.

Reviewer 1 ·

Basic reporting

The study is interesting, but I am not convinced by some of their images (esp. those in Fig. 1) and I found their presentation of data in Table 1 to be confusing. It was only after multiple reads and some back of the envelope calculations that I realized what they are showing.

It would really help the narrative if the authors could compare their findings to some metric of growth rate for these species. I'm not convinced about the Phoenix effect here because these are fast growing.

It would have been nice to see more specific data about colonies that suffered mortality but did not recover. I think these would have also been useful for comparison.

Figs. 2 and 4 are a little misleading as time periods where data are missing are shown the same as times when observations were made.

See below for more details.

Experimental design

Some minor additions to the design would be helpful to report: how many photos per colony were taken, what (more specifically) does ND mean, inclusion of some n-values would be helpful. See below for specific details.

Validity of the findings

I am struggling to see what the authors report as alive/dead in the images depicted in Fig. 1 for P. compressa. It is VERY difficult to see the turf algae and dead portions of the colony in the photos. I don’t know if this is something about the picture quality of my computer, but I am better able to see what they mean in Fig. 3 for M. capitata. Since I can see Fig. 3 distinctions better, I assume that Fig. 1 is more about photo quality, but this should be clarified by the authors. See below for more details.

Additional comments

The authors have assessed a suite of photographs of coral colonies from multiple years of observations following a bleaching event and have quantified coral mortality and recovery at fine-scale detail. Their study is based on a unique dataset that cannot be duplicated, is long-term (multi-year) and has interesting findings that may pertain to the Phoenix effect, the regrowth of coral skeleton from living tissue deep within the colony. The study is interesting, but I am not convinced by some of their images (esp. those in Fig. 1) and I found their presentation of data in Table 1 to be confusing. It was only after multiple reads and some back of the envelope calculations that I realized what they are showing. In addition, the conclusions drawn from the paper are relatively simplistic and not based on anything else but the % cover and recovery time data they have been determined. These species are incredibly well-studied and I think connections could be made to growth rates that would more clearly show if this is the Phoenix effect or just simply two fast growing species. In general, I think the data are worthy of publication, but the manuscript requires revisions before publication is warranted.
Major comments (in text order)
1. Tabular data is difficult to follow. Here are areas in the text/tables where the description and explanation of data can be more cleared explained:
a. Table 1: It took me several reads to realize that the 22 colonies that bleached for each species and reef are also counted within the total tagged value reported and not in addition to the other value. I had this same confusion for all rows of Table 1. I think it would be helpful to create separate bleached and non-bleached columns for each grouping and instead of including the total tagged colonies, include the quantities for each phenotypic status separately. I think it will be easier to follow the relevant text and percentages that are reported. Readers can then more clearly see that only bleached corals with partial mortality recovered. If there is a reason to report the data as is, then the table caption needs to more clearly indicate that the first value in each cell is the total number of colonies for that category and that the bleached value is also included in the total.
b. If Table 1 is edited as suggested or similar, this would really help the reading of the results where the numbers reported on L181, L193 would make more sense. For example, on my first reading, I thought that the value on L181 should be 32/163, not 19/119.
c. It would help the reader if appropriate n-values are included in the narrative of the methods that correspond to the details in Table 1. For examples, the n of unpaired non-bleached colonies could be included (L137-138); the n of the candidate colonies (L146) could be included.

2. Table 2 was easier to understand, but I think there is a missed opportunity to report the data as a rate. When you do this (max % mortality/months), there are some similarities among the rates. Colonies that bleached mildly in 2019 recovered at slower rate (1.4-1.5% per month) compared to colonies that didn’t bleach or were not observed in 2019 (faster rates of 2.6-7.5% per month). I’m not suggesting that they use this exact metric, but if there is some appropriate way to better combine the extent of mortality and recovery time, I think that would be helpful and more interesting. The data are currently just described in a one-dimensional way and I’m not sure there is much that can be said.

3. What is the growth rate of these species? How do you know that recover is “rapid” (L236)? There is no comparison to any other metric that could assess the pace of recovery. We’ve reported calcification rates of P. compressa as 40-80 mg/day when healthy and 0-60 mg/day when bleached. In M. capitata we reported very similar values for non-bleached and bleached colonies, respectively. See Rodrigues and Grottoli, 2006 for details. The authors can consider whether the data they have (cm2 per month) is comparable to that publication or to other more recent assessments of lateral growth and/or calcification (I’m not suggesting a self-citation here, esp. if the authors think this is not helpful). My point is that these are very fast growing species, even when bleached (although there’s a wider variability for sure when bleached). How do you assess the Phoenix effect in weedy species. I don’t think it’s appropriate to compare to these two species to massive Porites colonies with slow growth rates (see L63-65 and elsewhere in the paper). If the authors could compare their data to growth rates for these species, I think the story would be more compelling. As of now, these metrics of recovery are not compared to anything that is species relevant.

4. How often were photos taken for each colony? I think some more information about this aspect of the design would be helpful to include in the methods section.

5. What does the ND mean in Tables 1 and 2 beyond that you don’t have the data? Were these colonies not observed post-bleaching? If so, are there other data sets you can draw on, it seems strange to me to include Patch Reef 4 with so little data, especially when some comparisons are made between the two reefs presented.

6. The discussion about size of the colonies in the study (L238-241) is confusing to me as the authors report that size did not correlate with recovery time (L230-232). This part of the discussion seems to ignore the statistics.

7. I wish that the authors had included the data for the colonies that suffered mortality, but did not recovery to 90% of their tissue cover. From the narrative about the Matsuda paper (L283-292) it sounds like the details the current authors were able to measure are different from some of those past findings. I think it would help for comparison purposes to include information on these other colonies. I suspect in order to identify the 9 that they report here, they would have the data already for the other 228. Even reporting just a subset of those could be interesting. The authors make a really strong point (L137-138) that partial mortality of the unpaired non-bleached corals are reported for the first time, but they aren’t really discussed and no details are provided about these colonies except for what can be gleaned from Table 1. I think more info on these other colonies would help story.

8. I am struggling to see what the authors report as alive/dead in the images depicted in Fig. 1 for P. compressa. It is VERY difficult to see the turf algae and dead portions of the colony in the photos. I don’t know if this is something about the picture quality of my computer, but I am better able to see what they mean in Fig. 3 for M. capitata. Since I can see Fig. 3 distinctions better, I assume that Fig. 1 is more about photo quality, but this should be clarified by the authors. In Fig. 1, the differences that are depicted look like they could easily be bleached/non-bleached difference or shadows/dappling from the surface water.

9. Figs 2 and 4: The way the graphs are drawn and shaded is a little misleading since there is missing data for some colonies after 2019. I think the info report on L217-219 (for Fig. 2) and L225-227 (for Fig. 4) should be included as greyed or whited out areas on the relevant graphs.

Other comments to address
L32-33: Refers to only colonies that bleached in 2015 exhibited rapid tissue recovery. I agree that this is interesting, but what is this compared to? All the colonies reported in the manuscript bleached in 2015. I don’t see other years of data with % cover, mortality, or recovery reported.
L53: Both species in the study feed regularly when bleached and not bleached. I do not think that qualifies as an obligate symbiosis.
L213: Mild paling is considered 20% bleached by your own definitions. The tone here downplays this bleaching and I’m not sure why.
L312-315: Awkward statement and difficult to discern meaning. Do the authors mean that the turf algae is triggering the recolonization? I’m not sure what is being said or what evidence is used to support the statement.
L340-342: The authors pose a question in the conclusion, but I think they have omitted a possibility that there data and earlier discussion suggest. Earlier in the discussion they describe how environmental differences between the two patch reefs could be driving the patterns observed.
Fig. 3: Add a figure inset to show the colors that belong to alive/dead as done with Fig. 1.
Minor comments
L34: insert “previously” before “recovered”
L41-43: delete first sentence of intro, I think it is too broad and not needed
L74: I do not think the qualifier “hermatypic” is needed since all of these species qualify as hermatypic corals (reef builders). The exception could be Lobactis scutaria, but it is also a calcifier and contributes to reef building, if not actual construction. I think it is more clear to delete “hermatypic.”
L171: Font of “≥ 90%” is different from the rest
L185 and L195: delete both instances of “just”
L189: insert “13” after “reef”
L201: typographical error “(20200” should be “(2020)”
L204: change “between” to “among” since you are comparing more than just two colonies
L204: delete “For example” and begin a new paragraph with “The maximum…”
L219: begin new paragraph with “The maximum…”
L228: begin new paragraph with “Overall…”
L270: change “being” to “were”
L299: delete “Indeed” begin the sentence with “Both…”
L340: insert space between the words “for this”
Table 1 caption: delete sentences in lines 4-7, just keep “ND, no data”

·

Basic reporting

In the present manuscript, Jones and Barott provide evidence (with very few colonies) of the coral Phoenix effect or living tissue regrowing over dead colony parts of the Hawaiian corals Porites compressa and Montipora capitata after bleaching impacts.

While the manuscript is very well written and structured, my main concern is the very few colonies followed n = 9, with 7 P. compressa and 2 M. capitata, and the minor importance to the rest of the initial colonies tagged n = 237, between both species. The problem is that the science of the other colonies, as well as other important necessary data to this MS (e.g., comparison of accumulated heat stress across heatwave events, bleaching prevalences, genetics), are already published in other papers (e.g., Brown et al. 2023: https://doi.org/10.1073/pnas.2312104120, Matsuda et al. 2020: https://doi.org/10.3389/fevo.2020.00178). Accordingly, this manuscript feels like missing pieces of evidence, and to fully understand the story, readers need to go to these published manuscripts constantly. Yet, although the originality of this manuscript could be discussed, it is important to produce science to the fullest/end of every project, so the author’s diversification in smaller studies is remarkable.

Another important point is that it would be great to provide the amount of dead tissue and recovery tissue for the 9 colonies. There is a correlation figure between size and recovery, but more information is needed. Do you have time-lapses of how this recovery tissue occurred? Gradually from the skeleton, perhaps sites, suddenly?

Finally, the outcomes of the study, especially the title, are too powerful given that the study is only based on 9 colonies and with the two specific criteria to be accounted for the PE monitor (Lines 138-141). What happened to the other colonies (and they are the majority) is explained in other published papers. This should be further acknowledged in the manuscript and title. We need positive stories within the coral reef science world, but we need to remain neutral or purely objective in our statements. The title could be something like: “Despite XXX mortality, nine colonies evidence the Phoenix effect …”

Despite all these important issues raised, and if the paper becomes less powerful in certain statements, the content of the manuscript is probably worth publishing. I have provided some specific comments, hoping they will help improve the manuscript.

Experimental design

No comment.

Validity of the findings

No comment.

Additional comments

Specific comments:

Lines 15-37: The abstract needs the two criteria to be considered a colony accounting for the PE.

Line 15: Thermal coral bleaching is a phenomenon… Bleaching can occur for other stress reasons.

Line 19: The coral Phoenix effect…

Lines 19-21: This section is confusing. Do you mean that out of the 16% and 32%, half remained pigment until suffering mortality? Or the remaining pigmented colonies are the other % that did not suffer bleaching? Please clarify.

Line 33: Was the second heatwave stronger or weaker? You provide the results later in the discussion based on Brown et al. 2023. Please specify.

Lines 66-68: If the editorial accepts references in the abstract, I will acknowledge the authors who first described the PE in the abstract.

Line 70: “is a leading hypothesis”. Are there any non-leading hypotheses worth mentioning? Could this living tissue come from neighbouring living areas?
Lines 83-83: Please clarify what you mean by refuge populations. The PE is a high recovery resilience to overgrow over the recently dead tissues of the coral, but it is not a refuge population.

Line 112: You did not investigate the recovery dynamics of 237 colonies. You investigated the colonies with substantial partial mortality, specifically the 9 colonies showing recovery. Your study is a follow-up from other papers (which is okay), and this should be acknowledged here. You have all this information between lines 131-137.

Line 114: Please delete “routes of”

Line 115: Please delete “extreme”. The number of colonies experiencing mortality, according to Table 1, is not even ½. In scientific writing, it is always better to remain objective and neutral.

Lines 123-127: At what depths were patch reefs 4 and 13? In the discussion, you mention how some other stressors might explain the difference in your results. Further metadata to understand comparisons across patch reefs results are necessary to be displayed in the methods (e.g., back reef, distance to shore).

Lines 138-141: These two criteria are critical and should be included in the abstract and even briefly mentioned in the last section of the introduction.

Line 179: Based on your following sentence (19/119) I would replace “A large number” with “some” or go straight to sentence two.

Lines 186-187: 7 colonies out of the 19 colonies exhibiting recovery and out of the 119 tagged colonies. It does not seem like a lot for the strength of the title. Lines 193-194 also provide evidence that the title is too strong.

Lines 228-232: Did you test the mixed effects of the maximum extent of mortality and size (together) with recovery time?
Line 235: “Replace” several with “Nine” individuals

Lines 238-241: Do you mean the large area of the colony or the large area of the mortality patches? Table 2 provides % of mortality, but having an idea of the recovered dimensions will be useful. In the end, 57% of a 170 cm diameter colony is tricky to visualise without knowing the height of the colony.

Lines 242-243: Is there no hypothesis stating that the PE might come from the overgrowth of the neighbouring living tissue? When scientists make drills on massive Porites, the living tissue covers the hole regardless of how far it is underneath the skeleton hole. In many cases over 60 cm or more.

Lines 247-248: The comparing values of DHWs of 252 are necessary here to understand this sentence alone.

Lines 258-259: You state “the exception”, another sentence supporting why the title should change a bit. Perhaps: “Exceptional evidence of the Phoenix effect…”

Lines 259-265: This section needs clarification. If they maintained their pigmentation, how do you know if they died of bleaching? They might have died from something else.

Lines 269-283: This is a lovely discussion section, yet the methods need more metadata.

Line 330: Please replace “at least some individuals” with “nine individuals”.

Table 1: I do not understand the meaning of the () in the second row. Is it 11 with partial mortality and (6) out of 11 showing bleaching? The values of bleaching prevalence or the reference to the associated papers should be mentioned in the table or caption.

Table 2: In row Maximum mortality (%), do you mean 11% of the colony was dead? Please be more specific in the title. With just colony diameter, it is difficult to visualise the affected colony.

General comment for figures: Figures 1 and 3 are very similar. Figures 2 and 4 are also very similar. Please consider reducing the number of figures or being more creative in displaying figures differently.

---

## Round 0.2 · accepted · Accept

I have received an evaluation of your resubmission and am delighted to let you know that your manuscript is now suitable for publication in PeerJ. Thank you for attending to all of the comments made by the reviewers. Congratulations.

·

Basic reporting

The authors addressed both reviewer reports very well, and I do not have further reviewing comments.

Experimental design

No additional reviews of the experimental design.

Validity of the findings

No additional reviews of the findings.

Additional comments

No additional comments.